# Association between the Quantity of Nurse–Doctor Interprofessional Collaboration and in-Patient Mortality: A Systematic Review

**DOI:** 10.3390/ijerph21040494

**Published:** 2024-04-17

**Authors:** Sandesh Pantha, Martin Jones, Nompilo Moyo, Bijaya Pokhrel, Diana Kushemererwa, Richard Gray

**Affiliations:** 1School of Nursing and Midwifery, La Trobe University, Bundoora, VIC 3086, Australia; martin.jones@unisa.edu.au (M.J.); n.moyo@latrobe.edu.au (N.M.); b.pokhrel@latrobe.edu.au (B.P.); d.kushemererwa@latrobe.edu.au (D.K.); r.gray@latrobe.edu.au (R.G.); 2Department of Rural Health, University of South Australia, Whyalla Norrie, SA 5608, Australia; 3Victorian Tuberculosis Program, Melbourne Health, Melbourne, VIC 3000, Australia

**Keywords:** nurse–doctor collaboration, inpatient mortality, medical and surgical wards, systematic review

## Abstract

The level of nurse–doctor interprofessional collaboration may influence patient outcomes, including mortality. To date, no systematic reviews have investigated the association between the quantity of nurse–doctor interprofessional collaboration and inpatient mortality. A systematic review was conducted. We included studies that measured the quantity of nurse–doctor interprofessional collaboration and in-patient mortality. Five databases (MEDLINE, EMBASE, PsycINFO, CINAHL, and the Cochrane Register) were searched. Two researchers undertook the title, abstract, and full-text screening. The risk of bias was determined using the Effective Public Health Practice Project (EPHPP) critical appraisal tool. Six reports from three observational studies met the inclusion criteria. Participants included 1.32 million patients, 29,591 nurses, and 191 doctors. The included studies had a high risk of bias. Of the three studies, one reported a significant association and one found no association between the quantity of nurse–doctor collaboration and mortality. The third study reported on the quantity of nurse–doctor collaboration but did not report the test of this association. We found no high-quality evidence to suggest the amount of nurse–doctor interprofessional collaboration was associated with mortality in medical and surgical inpatients. There is a need for further high-quality research to evaluate the association between the amount of nurse–doctor collaboration and patient outcomes.

## 1. Background

Interprofessional collaboration in clinical settings refers to health professionals from different disciplines working together and sharing ideas and opinions to provide evidence-based patient care [1,2,3]. In most health services, the bulk of clinical care and decision-making is undertaken by nurses and doctors, making effective communication potentially critical to the provision of high-quality care [4]. It is unclear if the quantity and quality of nurse–doctor collaboration impact patient outcomes [5]. For example, it could be that poor collaboration can lead to a delay in sharing relevant clinical information, resulting in a delay in patient care [6]. Conversely, effective collaborative working may ensure patients receive timely evidence-based, informed care and treatment [7].

Existing systematic reviews that have examined the association between the quality of interprofessional collaboration and patient outcomes have reported inconsistent outcomes [8,9,10,11,12]. For example, Specchia et al. (2020) reported an umbrella review (a review of systematic reviews) to evaluate the effect of the quality of the interprofessional collaboration approach (team tumour board) on the management of cancer patients [8]. The authors concluded that better collaboration led to more timely diagnoses and lower rates of mortality. All included reviews were rated as having a high risk of bias when appraised against the AMSTAR2 (Assessing the Methodological Quality of Systematic Reviews) critical appraisal tool. Conversely, a systematic review by Reeves et al. (2017) of nine clinical trials involving 6540 participants reported no evidence of a significant association between the quality of interprofessional collaboration and patient outcomes [12]. Two other reviews have specifically focused on the association between the quality of nurse–doctor collaboration and patient outcomes [9,11]. A systematic review and meta-analysis of 30 randomised controlled trials involving 66,548 patients evaluating the effect of nurse–doctor interprofessional teamwork on patient outcomes in medical wards was reported by Pannick et al. (2015) [9]. The authors reported an eight percent reduction in the relative risk of hospital mortality in patients treated by a nurse–doctor interprofessional team compared to usual care. The included studies had a high risk of bias. Martin et al. (2010) reported a systematic review of 14 clinical trials that included 5530 patients [11]. A narrative synthesis identified one study where there was evidence that the quality of clinical collaboration was associated with patient mortality [11].

To date, no systematic reviews have examined the association between the quantity of interprofessional collaboration and inpatient mortality or any other patient outcomes. We identified two American multicentre observational studies where authors tested the association between the quantity of nurse–doctor collaboration and patient outcomes in medical and surgical wards [13,14]. Ma et al. (2015) reported that an increase in the quantity of nurse–doctor collaboration—determined using the Practice Environment Scale of the Nurse Work Index—was associated with a three percent reduction in 30-day hospital readmissions [13]. The association between the quantity of nurse–doctor collaboration—measured using the nurse–physician interaction scale—and the rate of hospital-acquired pressure ulcers in adult medical and surgical patients was investigated by Ma et al. (2018) [14]. The authors reported a 19% reduction in the odds of developing pressure ulcers if patients were admitted to a hospital ward with a high quantity of nurse–doctor collaboration [14].

The aim of this review is to investigate the association between the quantity of nurse–doctor collaboration and inpatient mortality.

## 2. Methods

We registered our review protocol on the Prospective Register of Systematic Reviews (PROSPERO Registration—CRD42019133543) before conducting our initial searches. We have published our protocol describing the review methodology in detail [15]. Our reporting complies with the updated Preferred Items for Systematic Review and Meta-Analysis (PRISMA) guideline, and we have included a completed PRISMA 2020 checklist with the manuscript (Appendix A, PRISMA checklist) [16].

### 2.1. Eligibility Criteria

We included observational and experimental studies if (1) the study reported a measure of the quantity of nurse–doctor interprofessional collaboration, (2) fieldwork was conducted in medical or surgical inpatient wards, (3) mortality was reported as an outcome, and (4) the manuscript was in English. Qualitative studies were not included in the review as they do not measure the association between nurse–doctor collaboration and patient outcomes.

### 2.2. Information Sources

Five databases were searched in this review: MEDLINE, EMBASE, PsycINFO, CINAHL, and the Cochrane Register. The Ovid platform was used to search the MEDLINE, EMBASE, and PsycINFO databases. We accessed CINAHL through Ebscohost. The initial search was carried out on 15 June 2019. The search was rerun twice—on 16 February 2021 and 23 May 2023—to identify any new studies that may have been published.

### 2.3. Search Strategy

We developed our search strategy around three concepts: ‘clinicians’ (nurses and doctors), ‘collaboration’, and ‘mortality’. Each concept was elaborated on using synonyms and truncations. For example, ‘cooperation’ and ‘coordination’ were used as synonyms for ‘collaboration’ as they are often used interchangeably [10,12]. As an example of a truncation used to identify clinicians, we used ‘physician*’. Our search strategy was initially co-designed with one information scientist and then checked by a second to ensure the rigor of our search strategies. We restricted our search to titles and abstracts to ensure that the number of citations generated was manageable. A complete search strategy for each database is available as a supplementary document (Appendix A, Search strategy).

### 2.4. Selection Process

Citations from individual databases were combined into a single endnote file (.xml). The .xml file was then uploaded into the ‘Covidence’ review manager software [17]. Duplicate citations were identified and removed in Covidence. Title, abstract, and full-text screening of eligible studies (against the inclusion criteria) was conducted by two researchers independently using Covidence. Any discrepancies between authors were resolved either by discussion or in consultation with a third member of the team.

### 2.5. Multiple Papers from a Single Study

If multiple papers were identified from a single study, we considered the first published paper reporting results from a study as the primary citation [18]. Information from both primary and secondary papers was extracted and compared. Where discrepancies were identified, we contacted the corresponding author to request clarification.

### 2.6. Checking for Retraction

We checked the paper’s entry on the journal website and the retraction watch database to confirm the manuscript had not been retracted.

### 2.7. Data Collection Process

We developed a template to support data extraction that was conducted independently by two reviewers (S.P. and B.P.). Any discrepancies in the data extraction were resolved by discussion between reviewers and checking information from the source manuscript. The corresponding author was contacted by email if additional information about the included study was required.

### 2.8. Data Items

We extracted the following information from the included papers: citation, year of publication, country where fieldwork was conducted, study setting, study design, sample size (number of patients, nurses, and doctors enrolled in the study), quantity of collaboration (and how it was determined), and mortality outcomes.

### 2.9. Study Risk of Bias Assessment

A quality appraisal of the included studies was undertaken using the Effective Public Health Practice Project (EPHPP) measure. The EPHPP has good psychometric properties and is applicable for both interventional and observational studies [19,20]. The EPHPP evaluates the risk of bias against six items: selection bias, study design, confounders, blinding, data collection method, and withdrawals/dropouts. Each of these categories is rated ‘strong’, ‘moderate’, or ‘weak’ against the criteria. Based on the number of weak ratings, an overall score is given: ‘strong’ (no weak ratings), ‘moderate’ (one weak rating), and ‘weak’ (two or more weak ratings). The two additional components of the quality appraisal (intervention integrity and analysis) do not contribute to the global rating.

Two authors (S.P. and B.P.) independently carried out the risk of bias assessment. Any discrepancies were resolved by discussion between the researchers.

### 2.10. Protocol Amendment

We amended the study protocol on 22 September 2020, clarifying that we would include PhD theses in the review.

## 3. Results

### 3.1. Study Selection

Figure 1 (PRISMA flowchart) shows the flow of studies through the review. Our initial search generated 23,159 citations.

The two repeat searches identified an additional 4070 and 5922 citations, respectively. Six reports from three discrete observational studies met our inclusion criteria. Five reports (three peer-reviewed manuscripts and two doctoral theses) were identified from the database search [21,22,23,24,25]. We obtained one additional manuscript through correspondence with the study author [26]. None of the studies had been retracted or corrected as of 12 March 2024.

### 3.2. Study Characteristics

Table 1 shows the characteristics of the three included studies. The fieldwork for all included studies was conducted in America using a cross-sectional observational design [21,23,24]. Of these three studies, two were conducted as part of the doctoral research [21,24]. Two authors collected data prospectively [21,23]. One was a secondary analysis of an existing dataset [24].

### 3.3. Clinical Settings

Two studies were conducted in the intensive care unit (ICU) [23,25] and the third in the surgical wards (acute care units) [24]. The Baggs (1990) study was carried out in a 17-bed medical ICU [21]. Three ICUs—two specialist units (medical and surgical) and one general unit (admitting both medical and surgical patients)—were involved in the Baggs et al. (1999) study [23]. The Kang (2016) study involved surgical wards from 665 hospitals in four geographical regions (California, Florida, New Jersey, and Pennsylvania) [24].

### 3.4. Study Participants

Participants included 191 doctors, 29,509 nurses, and 1.32 million patients (predominantly from the Kang (2016) study [24]). Kang (2016) did not involve doctors as study participants [24].

### 3.5. Definition of Mortality in the Included Studies

Mortality was defined by Baggs (1990) [21] and Baggs et al. (1999) [23] as death while in the hospital within one month of discharge from the ICU. In the Kang (2016) study, mortality was defined as death while in hospital or at home within 30 days of hospital admission [24].

### 3.6. Description of the Measures of the Quantity of Nurse–Doctor Collaboration

Three different self-administered measures were used to determine the level of nurse–doctor collaboration: the Decision About Transfer (DAT) scale [21], the Collaboration and Satisfaction About Care Decisions (CSACD) scale [27], and the collegial nurse–physician relations subscale of the Practice Environment Scale of the Nurse Work Index (PES-NWI) [28]. The DAT and CSACD were completed by both nurses and doctors. The PES-NWI was rated by nurses alone. Baggs (1990) [21] and Baggs et al. (1999) [23] measured the quantity of nurse–doctor collaboration at an individual patient level. The quantity of collaboration was reported at the hospital level in the Kang (2016) study [24]. A description of the measures of nurse–doctor collaboration used by the study authors is provided as a supplementary document (Appendix A, Characteristics of the measure of collaboration).

### 3.7. Decision about Transfer (DAT)

Baggs (1990) developed and used the Decision About Transfer (DAT) scale to measure the quantity of nurse–doctor collaboration [21]. The DAT is a measure of a decision about discharging patients from the ICU. Individual items are measured on a seven-point Likert scale ranging from ‘no collaboration’ to ‘maximum collaboration’. One item (item 7, ‘How much collaboration between nurses and doctors (physicians) occurred in making the transfer decision?’) measured the overall quantity of collaboration between nurses and doctors. The psychometric properties of the DAT score (construct validity and reliability) were established using the correlation between the DAT measure and two established measures of the quantity of nurse–doctor collaboration: the Collaborative Practice Scale (CPS) and the Index of Work Satisfaction (IWS). The authors reported a low to moderate correlation between DAT with CPS (r = 0.27, *p* = <0.05) and IWS (r = 0.24, *p* = <0.05) [21].

### 3.8. Collaboration and Satisfaction about Care Decisions (CSACD)

The quantity of nurse–doctor collaboration was measured using the Collaboration and Satisfaction About Care Decisions (CSACD) scale in the Baggs et al. (1999) study. The measure was developed from DAT [29]. The CSACD has nine items. The first seven items measure the perceived quantity of nurse–doctor collaboration on clinical decisions for the care provided to patients admitted to intensive care units [27]. Each item is scored on a seven-point Likert scale. The scores for the seven items are summed to generate a score for the quantity of collaboration (range 7–49) [29]. The other two items measure the level of satisfaction with the process of collaboration.

The construct validity of the CSACD instrument was tested with 58 nurses and doctors working in a paediatric ICU [27]. The validity of the measure was reported to be good [27]. The global collaboration score was correlated with the scores for the six individual attributes of collaboration.

### 3.9. Practice Environment Scale of the Nurse Work Index (PES-NWI)

Kang (2016) used the Practice Environment Scale of the Nurse Work Index (PES-NWI) to measure the quantity of nurse–doctor collaboration [24]. The quantity of nurse–doctor collaboration was measured by three items under the subscale ‘collegial nurse–physician relationship’: teamwork, quality of working relationships, and the extent of nurse–doctor collaboration [28]. Each item was measured on a 4-point Likert scale. An average score calculated across the three items was considered the quantity of nurse–doctor collaboration.

The validity of the NWI has been extensively studied and shown to have good psychometric properties [28,29,30,31]. Kang (2016) reports a high correlation (0.7) between the three items. Individual items had very high face values on factor analysis (teamwork: 0.83, good working relationship: 0.76, and collaboration: 0.82) [24].

### 3.10. Quality Appraisal

Table 2 provides a summary of the quality appraisal of the included studies. All included studies were rated as methodologically weak. All studies were rated strong on data collection methods and weak on confounders. Baggs (1990) was rated strong for selection bias [21]. Detailed information on the quality appraisal is provided with the manuscript as a supplementary document (Appendix A, Quality appraisal of the included studies).

### 3.11. Results of Individual Studies

Baggs (1990) reports no association between nurse reports of the quantity of nurse–doctor collaboration and mortality (B = −0.25, t = −1.83, *p* = 0.068) [21]. The authors adjusted for the severity of the patient’s illness and doctors’ reports of the quantity of collaboration with nurses [21].

The association between the quantity of nurse–doctor collaboration and mortality was not reported by Baggs et al. (1999) [23]. The reported outcome was a composite measure combining both morality and ICU readmission. Results were reported for each of the three participating ICUs individually. Overall results were not reported [23]. We contacted the corresponding author to see if it was possible to access the data. In response, we were informed that the data had been destroyed.

Kang (2016) reported that the quantity of nurse–doctor collaboration was significantly associated with reduced patient mortality (OR = 0.90, 95% CI 0.89, 0.91, *p* < 0.001) [24]. The strength of the association decreased after adjusting for patient (e.g., gender, nature of surgery, and comorbid conditions) and hospital characteristics (e.g., nurse–patient ratio, number of beds, and general or teaching hospital), perceived levels of teamwork, and level of educational preparation of nurses (number of graduate nurses) (OR = 0.98, 95% CI 0.96, 0.99, *p* < 0.001) [24].

### 3.12. Meta-Analysis

We were unable to extract the necessary data to enable us to undertake a meaningful meta-analysis.

## 4. Discussion

Our systematic review investigated the association between the quantity of nurse–doctor interprofessional collaboration and mortality in medical and surgical patients. Three observational studies were included, with all having a high risk of bias. One included study reported a significant association, and one reported no association between the quantity of nurse–doctor collaboration and mortality. One included study did not report the association between the quantity of nurse–doctor collaboration and mortality. Therefore, we cannot draw any meaningful conclusion on the association between the amount of nurse–doctor interprofessional collaboration and patient deaths in medical and surgical wards.

Existing systematic reviews showed inconsistent results on the association between the quality of collaboration and mortality [8,9,10,11,12]. For example, Pannick et al. (2015) reported that the quality of nurse–doctor collaboration was associated with reduced mortality [9]. Conversely, there was no association between the quality of collaboration and mortality in a review undertaken by Martin et al. (2010). Our review is the first to examine if the quantity of nurse–doctor collaboration is associated with patient outcomes. Overarchingly, there is seemingly no clear link between either the quality or quantity of nurse–doctor collaboration and mortality.

Two systematic reviews have examined the instruments measuring interprofessional collaboration [32,33]. For example, the authors of a scoping review of measures of interprofessional collaboration in healthcare settings identified 29 measures to determine the amount of collaboration between health professionals, with the majority of them focusing on nurse–doctor communication [32]. Researchers have measured the quantity of interprofessional collaboration but have not extended the work to show how this impacts patient outcomes [34,35,36]. Only a few studies have attempted to measure the quantity of collaboration between nurses and doctors [13,14]. For example, Ma et al. (2015) examined the association between the quantity of nurse–doctor collaboration and hospital readmissions [13]. We acknowledge that it is difficult to accurately link collaboration and patient outcomes. For example, patients may die because of the severity of the illness, which may not be attributable to the level of collaboration. This may be more common in ICUs, where more severe patients are admitted. In addition, patients may die at home or end up being admitted to a different hospital (due to consequences of the disease process or complications associated with the treatment), which may not be captured by routinely obtained data [37,38]. Perhaps the lack of research to examine the association between the quantity of nurse–doctor collaboration and patient outcomes is partially explained by the methodological challenges in conducting such studies [10,12].

All the included studies were observational. Almost all the participants were from one study. Data collection in the included studies was conducted at least fifteen years ago (Kang (2016) analysed a secondary data set that was collected in 2007) [24]. Because of the recent technological advances in clinical communication, there may be changes in the definition of what we consider effective collaboration. For example, a substantial amount of clinical communication is carried out through electronic medical records and voice notifications rather than direct face-to-face communication. None of the included studies reported if the influence of indirect communication between nurses and doctors had been considered.

Three studies were included in this review. The Cochrane Handbook does not recommend a minimum number of studies that need to be included in the review [39]. The quality of a systematic review greatly depends on the rigour of the research rather than the number of included studies [40]. A review with no (empty) review or with few included studies suggests a potential gap in knowledge, highlighting the need for primary research to address the omission [41,42]. Therefore, there is a justification for further research into the association between both the quantity and quality of nurse doctor collaboration and patient outcomes.

There are several limitations to our review that warrant consideration. Our review was narrow in scope and focused on a single outcome (mortality). We did this because mortality is an important objective outcome that is routinely recorded [43,44,45]. Other patient outcomes, such as readmission rates, may not be accurately and consistently recorded; however, they should be considered in future reviews [37,38,46]. We did not search the grey literature; authors have argued that it is difficult to develop a grey literature search strategy that can be replicated [47]. However, it is possible that we may have missed potentially relevant studies. A priori, we decided to exclude studies that were reported as conference proceedings. During the full-text screening, we identified 12 papers that were part of conference proceedings. The conference proceedings identified from the review potentially measured the interprofessional teams. For example, Bosso et al. (2017) compared the outcomes of ICU patients when an evening round was added to the usual care [48]. Cogan and Romero-Ortuno (2013) reported the outcome of geriatrician-led multidisciplinary management of patients residing in aged care services [49]. We are unsure if the conference proceedings excluded in the review reported the levels of nurse–doctor collaboration, as we could not obtain the full text of these studies. Two studies that were not in English were also excluded during full-text screening and again may have generated valuable data. We limited our search to the title and abstract only to make the number of searches manageable. For example, the initial search of the MEDLINE database resulted in more than 27,000 citations, reduced to 6700 after limiting it to title and abstract. There is a potential that some of the studies may have been missed because of our adjustment to the search strategy.

## 5. Conclusions

The quantity and quality of nurse–doctor collaboration is generally considered important to providing high-quality patient care. Our review highlights an important gap in knowledge around the importance of interprofessional collaboration. Further research is required.

## Figures and Tables

**Figure 1 ijerph-21-00494-f001:**
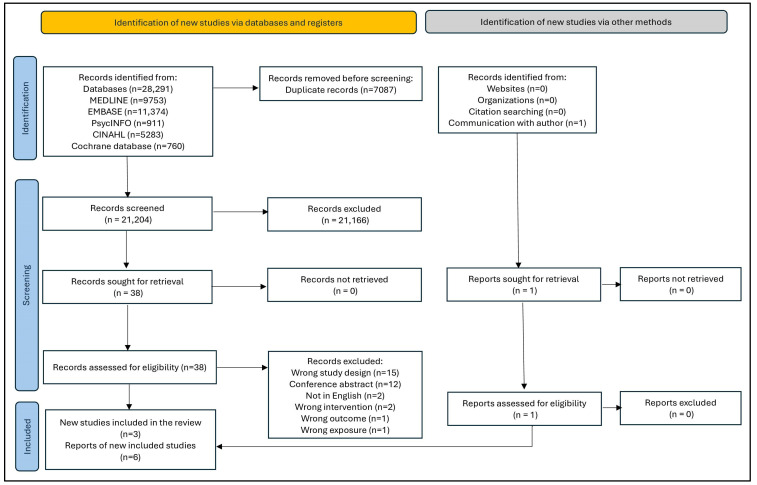
PRISMA flowchart ^1^ showing the number of manuscripts at each stage of the review. ^1^ One of the manuscripts was obtained through correspondence with the study author. This information has been reflected as a report sought for retrieval.

**Table 1 ijerph-21-00494-t001:** Study characteristics.

Citation	Additional Citation	Study Site	Study Type	Study Participants	Measure of the Quantity of Interprofessional Collaboration (Score Range)	Perceived Quantity of Nurse –Doctor Collaboration Mean (SD)	Outcome
Baggs (1990) ^1^ [21]	Baggs et al. (1992) [22]	17-bed medical ICU	prospective, observational	Patient (n = 286) Nurse (n = 56) Doctor (n = 31)	Decision About Transfer (1–7)	Nurse: 4.1 (2.2) Doctor: 4.4 (2.0)	Nurses’ reported collaboration not associated with mortality of patients (B = −0.25, t = −1.83, *p* = 0.068). A significant association between nurses’ reported collaboration and negative patient outcomes (B = −0.22, t = −2.34, *p* = 0.020). No association between doctors’ reported collaboration and negative patient outcomes.
Baggs et al. (1999) [23]	Baggs et al. ^2^ (1994) [26]	ICUs from three hospitals 16-bed medical ICU, 20-bed surgical ICU, 7-bed mixed ICU	prospective observational	Patient (n = 1432) Nurse (n = 162) Doctor (n = 160) [Resident doctor (n = 63), Attending doctor (n = 97)]	Collaboration and Satisfaction about Care Decisions (CSACD) scale (1–49)	*Medical ICU* ^3^ Nurse: 30.7 Doctor: 31.1 *Surgical ICU* ^3^ Nurse: 24.6 Resident: 27.8 Attending: 37.5 *Mixed ICU* ^3^ Nurse: 30.6 Doctor: 31.9	The association between collaboration and patient mortality was not reported in the manuscript. A significant association between nurses’ reported collaboration and negative patient outcomes (*p* = 0.037) in MICU. No association between nurses’ reported collaboration and negative patient outcomes in surgical and mixed ICUs. No statistically significant association between doctors’ reported collaboration and negative patient outcomes in any of the three ICUs.
Kang (2016) ^1^ [24]	Kang et al. (2020) [25]	Adult acute care hospitals (n = 665)	secondary data analysis	Patient (n = 1,321,904) Nurse (n = 29,391)	Nurse–physician relations subscale (1–4) in the Practice Environment Scale of the Nurse Work Index	Nurse: 2.90 (0.22)	A significant association between collaboration and patient outcomes [OR = 0.98, 95% CI = 0.96, 0.999, *p* < 0.001] even after controlling for patient and hospital characteristics, nurse teamwork, and education.

^1^ PhD dissertation. ^2^ The authors did not report the mean and standard deviation of CSACD scores of physicians and nurses in the manuscript. Information about the mean score was obtained from a separate manuscript reported from the same study. ^3^ Standard deviation was not reported.

**Table 2 ijerph-21-00494-t002:** Summary of quality appraisal using the Effective Public Health Practice Project (EPHPP) tool.

Scheme 1990	Criteria	Baggs (1990) [21]	Baggs et al. (1999) [23]	Kang (2016) [24]
A	Selection bias	Strong	Moderate	Moderate
B	Study design	Weak	Weak	Weak
C	Confounders	Weak	Weak	Weak
D	Blinding	Weak	Weak	Weak
E	Data collection methods	Strong	Strong	Strong
F	Withdrawals and drop-outs	Not applicable	Not applicable	Not applicable
G	Intervention integrity	-	-	-
H	Analysis	-	-	-
Global Rating 1 Strong (no weak ratings) 2 Moderate (one weak rating) 3 Weak (two or more weak ratings)		Weak	Weak	Weak

## Data Availability

Data sharing does not apply to this article as no datasets were generated or analysed during the current study.

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
