# Peer review of "Association between the Quantity of Nurse–Doctor Interprofessional Collaboration and in-Patient Mortality: A Systematic Review"

_ijerph, 2024, doi:10.3390/ijerph21040494_

Round 1

Reviewer 1 Report

Comments and Suggestions for Authors

Five databases (MEDLINE, EMBASE, PsycInfo, CINAHL, and the Cochrane Register) were not enough for this topic. The timeline is huge compared to the outcome. This systematic review is not truly able to express the real scenario of the nurse-doctor interprofessional relationship. In the last few years, I worked with some UK hospitals and universities on this topic, which published most of the low-impact, or Google Scholar, scopus based journals. There is a lot of research on this topic done in a qualitative manner. I think the whole method of search strategy is not thoroughly checked, which makes it necessary to recalibrate the search strategy and data selection procedure. 

**Also, keep in mind that nursing-related studies are not well-reported, and in recent years, we have encouraged them to submit their reports to journals.

Comments on the Quality of English Language

Authors need proofreading for syntax and grammar 

Author Response

Dear Reviewer, 

Thank you for your time and effort to review our manuscript. Please see the attached file for a point-by-point response.

Kind regards,

Sandesh Pantha

Reviewer 2 Report

Comments and Suggestions for Authors

The paper Association between the levels of nurse-doctor interprofessional collaboration and in-patient mortality: A systematic review, signed by 6 authors, investigate the level of collaboration between nurses and doctors with impact in patients’ mortality.

From methodological point of view the paper seems to be very well written, structured, and focus on details.

However, reading the article, I have noticed some inconvenients.

Thus,

1.     The references are inserted in the text, with names, and they should be inserted with numbers, according to the publication guidelines:  „In the text, reference numbers should be placed in square brackets [ ] and placed before the punctuation; for example [1], [1–3] or [1,3]. For embedded citations in the text with pagination, use both parentheses and brackets to indicate the reference number and page numbers; for example [5] (p. 10), or [6] (pp. 101–105)”. I recommend authors to reframe references.

2.     Some authors names are incorrect (Specchia), see lines 44 and 45.

3.     You mention different numbers of studies included in your review: 6 in abstract (‘Six documents from three observational studies met the inclusion criteria’; 5 in results (Five documents (three peer-reviewed manuscripts and two doctoral theses) from three discrete observational studies met our inclusion criteria”); and 3 in the table 1. I had some confusion on understanding why this different numbers. Maybe you can clarify us.

4.     In the limitation section, lines 300-303, you mention about the exclusion of 12 potentially relevant papers published as conference proceedings, reason: they ‘would weaken the rigour of our work’. I do not understand how they could do this. In my perception a higher number of studies could improve knowledge and make readers to understand your topic, the interaction between doctors and nurses with impact in patients’ mortality. Also, I feel like you ‘sacrificed’ the knowledge you could obtained by taking into consideration all materials and proving readers with interesting insights.

5.     In conclusion you mention about the need to conduct research in this topic. I wonder if you, as authors, get some benefits from doing this work: do you have enough information in order to conduct yourself research on the topic? 

Author Response

(The authors gave the same response as above.)

Reviewer 3 Report

Comments and Suggestions for Authors

1.            The entire document, including the abstract, requires editing by a native English speaker.

2.            What is meant by 'positive collaborative'? Could you cite relevant literature to clarify this term?

3.            The statement 'There have been four systematic and one umbrella review that have examined the effect of nurse and doctor teamwork - in which levels of collaboration is a component - on patient outcomes' is unclear.

4.            The frequent use of dashes from lines 61-66 disrupts the readability of the text. Need to revise this section for smoother flow.

5.            The sentence 'We included mortality as the only outcome of interest as mortality is one of the important hospital performance indicators and is valid' is ambiguous. Please clarify.

6.            The background section does not mention relevant literature regarding the outcome (inpatient mortality) being investigated. Please add and cite relevant literature.

7.            The section on 'Information sources' seems to omit data collection for the year 2020.please address this omission.

8.            The exclusion criteria are not mentioned, and the page numbers in your PRISMA 2020 Main Checklist do not match those in your article. Please rectify.

9.            While you state that 'Teamwork and inter-professional collaboration are related but different concepts, with collaboration being a core component of effective team working,' the search strategy in Supplementary material S2_Complete search strategy includes 'teamwork' as a keyword. Please reassess your keywords and MeSH terms.

10.         There is a typographical error on line 137: 'EHPPP'.

11.         Figure 1. PRISMA indicates one report sought for retrieval, but the previous process (record identified from) did not identify any literature.

12.         Figure 1. PRISMA's final process mentions reports of new included studies n=6, yet the document only discusses three articles.

13.         Overall, reviewing only three articles seems insufficient for a systematic literature review to be published.

Comments on the Quality of English Language

Extensive editing of English language required.

Author Response

(The authors gave the same response as above.)

Reviewer 4 Report

Comments and Suggestions for Authors

This review investigates the evidence around the level of nurse-doctor collaboration and inpatient mortality. It states that any conclusions cannot be drawn from previously conducted studies about this topic. The manuscript is well written following the PRISMA checklist with strict criteria. References numbers in the text are missing, authors used the textual form of noting references instead (this should be changed to numbers). Line 271 is the real conclusion of this review and I suggest that it should be transferred as the first sentence of the conclusion. Furthermore, “Limitations” should be placed in the “Discussion” section, not as separate.

In total, this study creates a good insight into the lack of evidence in nurse-doctor interprofessional collaboration and patient outcomes. It can be accepted for publication after the changes mentioned earlier are made. 

Author Response

(The authors gave the same response as above.)

Round 2

Reviewer 3 Report

Comments and Suggestions for Authors

your paper makes a valuable contribution to the field and, with these suggested revisions, has the potential to make an even more significant impact. I look forward to seeing the further development of your work and its eventual publication.